# Manifold-regression to predict from MEG/EEG brain signals without source modeling

**David Sabbagh** [*†‡], **Pierre Ablin, Gaël Varoquaux, Alexandre Gramfort, Denis A. Engemann** [§]
Université Paris-Saclay, Inria, CEA, Palaiseau, 91120, France

## Abstract

Magnetoencephalography and electroencephalography (M/EEG) can reveal neuronal dynamics non-invasively in real-time and are therefore appreciated methods in medicine and neuroscience. Recent advances in modeling brain-behavior relationships have highlighted the effectiveness of Riemannian geometry for summarizing the spatially correlated time-series from M/EEG in terms of their covariance. However, after artefact-suppression, M/EEG data is often rank deficient which limits the application of Riemannian concepts. In this article, we focus on the task of *regression* with rank-reduced covariance matrices. We study two Riemannian approaches that vectorize the M/EEG covariance between-sensors through projection into a tangent space. The Wasserstein distance readily applies to rank-reduced data but lacks affine-invariance. This can be overcome by finding a common subspace in which the covariance matrices are full rank, enabling the affine-invariant geometric distance. We investigated the implications of these two approaches in synthetic generative models, which allowed us to control estimation bias of a linear model for prediction. We show that Wasserstein and geometric distances allow perfect out-of-sample prediction on the generative models. We then evaluated the methods on real data with regard to their effectiveness in predicting age from M/EEG covariance matrices. The findings suggest that the data-driven Riemannian methods outperform different sensor-space estimators and that they get close to the performance of biophysics-driven source-localization model that requires MRI acquisitions and tedious data processing. Our study suggests that the proposed Riemannian methods can serve as fundamental building-blocks for automated large-scale analysis of M/EEG.

## 1 Introduction

Magnetoencephalography and electroencephalography (M/EEG) measure brain activity with millisecond precision from outside the head [23]. Both methods are non-invasive and expose rhythmic signals induced by coordinated neuronal firing with characteristic periodicity between minutes and milliseconds [10]. These so-called brain-rhythms can reveal cognitive processes as well as health status and are quantified in terms of the spatial distribution of the power spectrum over the sensor array that samples the electromagnetic fields around the head [3].

Statistical learning from M/EEG commonly relies on covariance matrices estimated from band-pass filtered signals to capture the characteristic scale of the neuronal events of interest [7, 22, 16]. However, covariance matrices do not live in an Euclidean space but a Riemannian manifold.

---

[*] Additional affiliation: Inserm, UMRS-942, Paris Diderot University, Paris, France

[†] Additional affiliation: Department of Anaesthesiology and Critical Care, Lariboisière Hospital, Assistance Publique Hôpitaux de Paris, Paris, France

[‡] dav.sabbagh@gmail.com

[§] denis-alexander.engemann@inria.fr

Fortunately, Riemannian geometry offers a principled mathematical approach to use standard linear learning algorithms such as logistic or ridge regression that work with Euclidean geometry. This is achieved by projecting the covariance matrices into a vector space equipped with an Euclidean metric, the tangent space. The projection is defined by the Riemannian metric, for example the geometric affine-invariant metric [5] or the Wasserstein metric [6]. As a result, the prediction error can be substantially reduced when learning from covariance matrices using Riemannian methods [45, 14].

In practice, M/EEG data is often provided in a rank deficient form by platform operators but also curators of public datasets [32, 2]. Its contamination with high-amplitude environmental electromagnetic artefacts often render aggressive offline-processing mandatory to yield intelligible signals. Commonly used tools for artefact-suppression project the signal linearly into a lower dimensional subspace that is hoped to predominantly contain brain signals [40, 42, 34]. But this necessarily leads to inherently rank-deficient covariance matrices for which no affine-invariant distance is defined. One remedy may consist in using anatomically informed source localization techniques that can typically deal with rank deficiencies [17] and can be combined with source-level estimators of neuronal interactions [31]. However, such approaches require domain-specific expert knowledge, imply processing steps that are hard to automate (e.g. anatomical coregistration) and yields pipelines in which excessive amounts of preprocessing are not under control of the predictive model.

In this work, we focus on regression with rank-reduced covariance matrices. We propose two Riemannian methods for this problem. A first approach uses a Wasserstein metric that can handle rank-reduced matrices, yet is not affine-invariant. In a second approach, matrices are projected into a common subspace in which affine-invariance can be provided. We show that both metrics can achieve perfect out-of-sample predictions in a synthetic generative model. Based on the SPoC method [15], we then present a supervised and computationally efficient approach to learn subspace projections informed by the target variable. Finally, we apply these models to the problem of inferring age from brain data [33, 31] on 595 MEG recordings from the Cambridge Center of Aging (Cam-CAN, http://cam-can.org) covering an age range from 18 to 88 years [41]. We compare the data-driven Riemannian approaches to simpler methods that extract power estimates from the diagonal of the sensor-level covariance as well as the cortically constrained minimum norm estimates (MNE) which we use to project the covariance into a subspace defined by anatomical prior knowledge.

**Notations** We denote scalars $s \in \mathbb{R}$ with regular lowercase font, vectors $\boldsymbol{s} = [s_1, \ldots, s_N] \in \mathbb{R}^N$ with bold lowercase font and matrices $\boldsymbol{S} \in \mathbb{R}^{N \times M}$ with bold uppercase fonts. $\boldsymbol{I}_N$ is the identity matrix of size $N$. $[\cdot]^\top$ represents vector or matrix transposition. The Frobenius norm of a matrix will be denoted by $||\boldsymbol{M}||_F^2 = \text{Tr}(\boldsymbol{M}\boldsymbol{M}^\top) = \sum |M_{ij}|^2$ with $\text{Tr}(\cdot)$ the trace operator. $\text{rank}(\boldsymbol{M})$ is the rank of a matrix. The $l_2$ norm of a vector $\boldsymbol{x}$ is denoted by $||\boldsymbol{x}||_2^2 = \sum x_i^2$. We denote by $\mathcal{M}_P$ the space of $P \times P$ square real-valued matrices, $\mathcal{S}_P = \{\boldsymbol{M} \in \mathcal{M}_P, \boldsymbol{M}^\top = \boldsymbol{M}\}$ the subspace of symmetric matrices, $\mathcal{S}_P^{++} = \{\boldsymbol{S} \in \mathcal{S}_P, \boldsymbol{x}^\top S \boldsymbol{x} > 0, \forall \boldsymbol{x} \in \mathbb{R}^P\}$ the subspace of $P \times P$ symmetric positive definite matrices, $\mathcal{S}_P^+ = \{\boldsymbol{S} \in \mathcal{S}_P, \boldsymbol{x}^\top S \boldsymbol{x} \geq 0, \forall \boldsymbol{x} \in \mathbb{R}^P\}$ the subspace of $P \times P$ symmetric semi-definite positive (SPD) matrices, $\mathcal{S}_{P,R}^+ = \{\boldsymbol{S} \in \mathcal{S}_P^+, \text{rank}(\boldsymbol{S}) = R\}$ the subspace of SPD matrices of fixed rank R. All matrices $\boldsymbol{S} \in \mathcal{S}_P^{++}$ are full rank, invertible (with $\boldsymbol{S}^{-1} \in \mathcal{S}_P^{++}$) and diagonalizable with real strictly positive eigenvalues: $\boldsymbol{S} = \boldsymbol{U}\boldsymbol{\Lambda}\boldsymbol{U}^\top$ with $\boldsymbol{U}$ an orthogonal matrix of eigenvectors of $\boldsymbol{S}$ ($\boldsymbol{U}\boldsymbol{U}^\top = \boldsymbol{I}_P$) and $\boldsymbol{\Lambda} = \text{diag}(\lambda_1, \ldots, \lambda_n)$ the diagonal matrix of its eigenvalues $\lambda_1 \geq \ldots \geq \lambda_n > 0$. For a matrix $\boldsymbol{M}$, $\text{diag}(\boldsymbol{M}) \in \mathbb{R}^P$ is its diagonal. We also define the exponential and logarithm of a matrix: $\forall \boldsymbol{S} \in \mathcal{S}_P^{++}, \log(\boldsymbol{S}) = \boldsymbol{U} \text{ diag}(\log(\lambda_1), \ldots, \log(\lambda_n)) \boldsymbol{U}^\top \in \mathcal{S}_P$, and $\forall \boldsymbol{M} \in \mathcal{S}_P, \exp(\boldsymbol{M}) = \boldsymbol{U} \text{ diag}(\exp(\lambda_1), \ldots, \exp(\lambda_n)) \boldsymbol{U}^\top \in \mathcal{S}_P^{++}$. $\mathcal{N}(\mu, \sigma^2)$ denotes the normal (Gaussian) distribution of mean $\mu$ and variance $\sigma^2$. Finally, $\mathbb{E}_s[\boldsymbol{x}]$ represents the expectation and $\mathbb{V}\text{ar}_s[\boldsymbol{x}]$ the variance of any random variable $\boldsymbol{x}$ w.r.t. their subscript $s$ when needed.

**Background and M/EEG generative model** MEG or EEG data measured on $P$ channels are multivariate signals $\boldsymbol{x}(t) \in \mathbb{R}^P$. For each subject $i = 1 \ldots N$, the data are a matrix $\boldsymbol{X}_i \in \mathbb{R}^{P \times T}$ where $T$ is the number of time samples. For the sake of simplicity, we assume that $T$ is the same for each subject, although it is not required by the following method. The *linear instantaneous mixing model* is a valid generative model for M/EEG data due to the linearity of Maxwell's equations [23]. Assuming the signal originates from $Q < P$ locations in the brain, at any time $t$, the measured signal

vector of subject $i = 1 \ldots N$ is a linear combination of the $Q$ *source patterns* $\boldsymbol{a}_j^s \in \mathbb{R}^P$, $j = 1 \ldots Q$:

$$\boldsymbol{x}_i(t) = \boldsymbol{A}^s \, \boldsymbol{s}_i(t) + \boldsymbol{n}_i(t) \ , \tag{1}$$

where the patterns form the time and subject-independent source *mixing matrix* $\boldsymbol{A}^s = [\boldsymbol{a}_1^s, \ldots, \boldsymbol{a}_Q^s] \in \mathbb{R}^{P \times Q}$, $\boldsymbol{s}_i(t) \in \mathbb{R}^Q$ is the *source vector* formed by the $Q$ time-dependent sources amplitude, $\boldsymbol{n}_i(t) \in \mathbb{R}^P$ is a contamination due to noise. Note that the mixing matrix $\boldsymbol{A}^s$ and sources $\boldsymbol{s}_i$ are not known.

Following numerous learning models on M/EEG [7, 15, 22], we consider a regression setting where the target $y_i$ is a function of the power of the sources, denoted $p_{i,j} = \mathbb{E}_t[s_{i,j}^2(t)]$. Here we consider the linear model:

$$y_i = \sum_{j=1}^{Q} \alpha_j f(p_{i,j}) \ , \tag{2}$$

where $\boldsymbol{\alpha} \in \mathbb{R}^Q$ and $f : \mathbb{R}^+ \to \mathbb{R}$ is increasing. Possible choices for $f$ that are relevant for neuroscience are $f(x) = x$, or $f(x) = \log(x)$ to account for log-linear relationships between brain signal power and cognition [7, 22, 11]. A first approach consists in estimating the sources before fitting such a linear model, for example using the Minimum Norm Estimator (MNE) approach [24]. This boils down to solving the so-called M/EEG inverse problem which requires costly MRI acquisitions and tedious processing [3]. A second approach is to work directly with the signals $\boldsymbol{X}_i$. To do so, models that enjoy some invariance property are desirable: these models are blind to the mixing $\boldsymbol{A}^s$ and working with the signals $\boldsymbol{x}$ is similar to working directly with the sources $\boldsymbol{s}$. Riemannian geometry is a natural setting where such invariance properties are found [18]. Besides, under Gaussian assumptions, model (1) is fully described by second order statistics [37]. This amounts to working with covariance matrices, $\boldsymbol{C}_i = \boldsymbol{X}_i \boldsymbol{X}_i^\top / T$, for which Riemannian geometry is well developed. One specificity of M/EEG data is, however, that signals used for learning have been rank-reduced. This leads to rank-deficient covariance matrices, $\boldsymbol{C}_i \in \mathcal{S}_{P,R}^+$, for which specific matrix manifolds need to be considered.

## 2 Theoretical background to model invariances on $\mathcal{S}_{P,R}^+$ manifold

### 2.1 Riemannian matrix manifolds

Endowing a continuous set $\mathcal{M}$ of square matrices with a metric, that defines a local Euclidean structure, gives a Riemannian manifold with a solid theoretical framework. Let $\boldsymbol{M} \in \mathcal{M}$, a $K$-dimensional Riemannian manifold. For any matrix $\boldsymbol{M}' \in \mathcal{M}$, as $\boldsymbol{M}' \to \boldsymbol{M}$, $\boldsymbol{\xi}_{\boldsymbol{M}} = \boldsymbol{M}' - \boldsymbol{M}$ belongs to a vector space $\mathcal{T}_{\boldsymbol{M}}$ of dimension $K$ called the *tangent space* at $\boldsymbol{M}$.

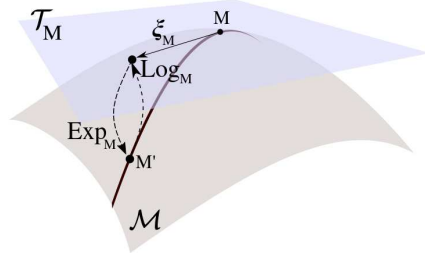

The *Riemannian metric* defines an inner product $\langle \cdot, \cdot \rangle_{\boldsymbol{M}} : \mathcal{T}_{\boldsymbol{M}} \times \mathcal{T}_{\boldsymbol{M}} \to \mathbb{R}$ for each tangent space $\mathcal{T}_{\boldsymbol{M}}$, and as a consequence a norm in the tangent space $\|\boldsymbol{\xi}\|_{\boldsymbol{M}} = \sqrt{\langle \boldsymbol{\xi}, \boldsymbol{\xi} \rangle_{\boldsymbol{M}}}$. Integrating this metric between two points gives a *geodesic* distance $d : \mathcal{M} \times \mathcal{M} \to \mathbb{R}^+$. It allows to define means on the manifold:

Figure 1: Tangent Space, exponential and logarithm on Riemannian manifold illustration.

$$\mathrm{Mean}_d(\boldsymbol{M}_1, \ldots, \boldsymbol{M}_N) = \arg \min_{\boldsymbol{M} \in \mathcal{M}} \sum_{i=1}^{N} d(\boldsymbol{M}_i, \boldsymbol{M})^2 \ . \tag{3}$$

The *manifold exponential* at $\boldsymbol{M} \in \mathcal{M}$, denoted $\mathrm{Exp}_{\boldsymbol{M}}$, is a smooth mapping from $T_{\boldsymbol{M}}$ to $\mathcal{M}$ that preserves local properties. In particular, $d(\mathrm{Exp}_{\boldsymbol{M}}(\boldsymbol{\xi}_{\boldsymbol{M}}), \boldsymbol{M}) = \|\boldsymbol{\xi}_{\boldsymbol{M}}\|_{\boldsymbol{M}}$ for $\boldsymbol{\xi}_{\boldsymbol{M}}$ small enough. Its inverse is the *manifold logarithm* $\mathrm{Log}_{\boldsymbol{M}}$ from $\mathcal{M}$ to $\mathcal{T}_{\boldsymbol{M}}$, with $\|\mathrm{Log}_{\boldsymbol{M}}(\boldsymbol{M}')\|_{\boldsymbol{M}} = d(\boldsymbol{M}, \boldsymbol{M}')$ for $\boldsymbol{M}, \boldsymbol{M}' \in \mathcal{M}$. Finally, since $\mathcal{T}_{\boldsymbol{M}}$ is Euclidean, there is a linear invertible mapping $\phi_{\boldsymbol{M}} : \mathcal{T}_{\boldsymbol{M}} \to \mathbb{R}^K$ such that for all $\xi_{\boldsymbol{M}} \in \mathcal{T}_{\boldsymbol{M}}$, $\|\boldsymbol{\xi}_{\boldsymbol{M}}\|_{\boldsymbol{M}} = \|\phi_{\boldsymbol{M}}(\boldsymbol{\xi}_{\boldsymbol{M}})\|_2$. This allows to define the *vectorization operator* at $\boldsymbol{M} \in \mathcal{M}$, $\mathcal{P}_{\boldsymbol{M}} : \mathcal{M} \to \mathbb{R}^K$, defined by $\mathcal{P}_{\boldsymbol{M}}(\boldsymbol{M}') = \phi_{\boldsymbol{M}}(\mathrm{Log}_{\boldsymbol{M}}(\boldsymbol{M}'))$. Fig. 1 illustrates these concepts.

The vectorization explicitly captures the local Euclidean properties of the Riemannian manifold:

$$d(\boldsymbol{M}, \boldsymbol{M}') = \|\mathcal{P}_{\boldsymbol{M}}(\boldsymbol{M}')\|_2 \tag{4}$$

Hence, if a set of matrices $\boldsymbol{M}_1, \ldots, \boldsymbol{M}_N$ is located in a small portion of the manifold, denoting $\overline{\boldsymbol{M}} = \text{Mean}_d(\boldsymbol{M}_1, \ldots, \boldsymbol{M}_N)$, it holds:

$$d(\boldsymbol{M}_i, \boldsymbol{M}_j) \simeq \|\mathcal{P}_{\overline{\boldsymbol{M}}}(\boldsymbol{M}_i) - \mathcal{P}_{\overline{\boldsymbol{M}}}(\boldsymbol{M}_j)\|_2 \tag{5}$$

For additional details on matrix manifolds, see [1], chap. 3.

**Regression on matrix manifolds**  The vectorization operator is key for machine learning applications: it projects points in $\mathcal{M}$ on $\mathbb{R}^K$, and the distance $d$ on $\mathcal{M}$ is approximated by the distance $\ell_2$ on $\mathbb{R}^K$. Therefore, those vectors can be used as input for any standard regression technique, which often assumes a Euclidean structure of the data. More specifically, throughout the article, we consider the following regression pipeline. Given a training set of samples $\boldsymbol{M}_1, \ldots, \boldsymbol{M}_N \in \mathcal{M}$ and target continuous variables $y_1, \ldots, y_N \in \mathbb{R}$, we first compute the mean of the samples $\overline{\boldsymbol{M}} = \text{Mean}_d(\boldsymbol{M}_1, \ldots, \boldsymbol{M}_N)$. This mean is taken as the reference to compute the vectorization. After computing $\boldsymbol{v}_1, \ldots, \boldsymbol{v}_N \in \mathbb{R}^K$ as $\boldsymbol{v}_i = \mathcal{P}_{\overline{\boldsymbol{M}}}(\boldsymbol{M}_i)$, a linear regression technique (e.g. ridge regression) with parameters $\boldsymbol{\beta} \in \mathbb{R}^K$ can be employed assuming that $y_i \simeq \boldsymbol{v}_i^\top \boldsymbol{\beta}$.

## 2.2 Distances and invariances on positive matrices manifolds

We will now introduce two important distances: the geometric distance on the manifold $\mathcal{S}_P^{++}$ (also known as affine-invariant distance), and the Wasserstein distance on the manifold $\mathcal{S}_{P,R}^+$.

**The geometric distance**  Seeking properties of covariance matrices that are invariant by linear transformation of the signal, leads to endow the positive definite manifold $S_P^{++}$ with the *geometric* distance [18]:

$$d_G(\boldsymbol{S}, \boldsymbol{S}') = \|\log(\boldsymbol{S}^{-1}\boldsymbol{S}')\|_F = \left[\sum_{i=1}^P \log^2 \lambda_k\right]^{\frac{1}{2}} \tag{6}$$

where $\lambda_k, k = 1 \ldots P$ are the real eigenvalues of $\boldsymbol{S}^{-1}\boldsymbol{S}'$. The affine invariance property writes:

$$\text{For } \boldsymbol{W} \text{ invertible}, d_G(\boldsymbol{W}^\top \boldsymbol{S}\boldsymbol{W}, \boldsymbol{W}^\top \boldsymbol{S}'\boldsymbol{W}) = d_G(\boldsymbol{S}, \boldsymbol{S}') \ . \tag{7}$$

This distance gives a Riemannian-manifold structure to $S_P^{++}$ with the inner product $\langle \boldsymbol{P}, \boldsymbol{Q} \rangle_{\boldsymbol{S}} = \text{Tr}(\boldsymbol{P}\boldsymbol{S}^{-1}\boldsymbol{Q}\boldsymbol{S}^{-1})$ [18]. The corresponding manifold logarithm at $\boldsymbol{S}$ is $\text{Log}_{\boldsymbol{S}}(\boldsymbol{S}') = \boldsymbol{S}^{\frac{1}{2}} \log\left(\boldsymbol{S}^{-\frac{1}{2}}\boldsymbol{S}'\boldsymbol{S}^{-\frac{1}{2}}\right)\boldsymbol{S}^{\frac{1}{2}}$ and the vectorization operator $\mathcal{P}_{\boldsymbol{S}}(\boldsymbol{S}')$ of $\boldsymbol{S}'$ *w.r.t.* $\boldsymbol{S}$: $\mathcal{P}_{\boldsymbol{S}}(\boldsymbol{S}') = \text{Upper}(\boldsymbol{S}^{-\frac{1}{2}}\text{Log}_{\boldsymbol{S}}(\boldsymbol{S}')\boldsymbol{S}^{-\frac{1}{2}}) = \text{Upper}(\log(\boldsymbol{S}^{-\frac{1}{2}}\boldsymbol{S}'\boldsymbol{S}^{-\frac{1}{2}}))$, where $\text{Upper}(\boldsymbol{M}) \in \mathbb{R}^K$ is the vectorized upper-triangular part of $\boldsymbol{M}$, with unit weights on the diagonal and $\sqrt{2}$ weights on the off-diagonal, and $K = P(P+1)/2$.

**The Wasserstein distance**  Unlike $\mathcal{S}_P^{++}$, it is hard to endow the $\mathcal{S}_{P,R}^+$ manifold with a distance that yields tractable or cheap-to-compute logarithms [43]. This manifold is classically viewed as $\mathcal{S}_{P,R}^+ = \{\mathbf{Y}\mathbf{Y}^\top | \mathbf{Y} \in \mathbb{R}_*^{P \times R}\}$, where $\mathbb{R}_*^{P \times R}$ is the set $P \times R$ matrices of rank $R$ [30]. This view allows to write $\mathcal{S}_{P,R}^+$ as a quotient manifold $\mathbb{R}_*^{P \times R}/\mathcal{O}_R$, where $\mathcal{O}_R$ is the orthogonal group of size $R$. This means that each matrix $\mathbf{Y}\mathbf{Y}^\top \in \mathcal{S}_{P,R}^+$ is identified with the set $\{\mathbf{Y}\mathbf{Q} | \mathbf{Q} \in \mathcal{O}_R\}$.

It has recently been proposed [35] to use the standard Frobenius metric on the total space $\mathbb{R}_*^{P \times R}$. This metric in the total space is equivalent to the *Wasserstein* distance [6] on $\mathcal{S}_{P,R}^+$:

$$d_W(\boldsymbol{S}, \boldsymbol{S}') = \left[\text{Tr}(\boldsymbol{S}) + \text{Tr}(\boldsymbol{S}') - 2\text{Tr}((\boldsymbol{S}^{\frac{1}{2}}\boldsymbol{S}'\boldsymbol{S}^{\frac{1}{2}})^{\frac{1}{2}})\right]^{\frac{1}{2}} \tag{8}$$

This provides cheap-to-compute logarithms:

$$\text{Log}_{\boldsymbol{Y}\boldsymbol{Y}^\top}(\boldsymbol{Y}'\boldsymbol{Y}'^\top) = \boldsymbol{Y}'\boldsymbol{Q}^* - \boldsymbol{Y} \in \mathbb{R}_*^{P \times R} \ , \tag{9}$$

where $\boldsymbol{U}\boldsymbol{\Sigma}\boldsymbol{V}^\top = \boldsymbol{Y}^\top\boldsymbol{Y}'$ is a singular value decomposition and $\boldsymbol{Q}^* = \boldsymbol{V}\boldsymbol{U}^\top$. The vectorization operator is then given by $\mathcal{P}_{\boldsymbol{Y}\boldsymbol{Y}^\top}(\boldsymbol{Y}'\boldsymbol{Y}'^\top) = \text{vect}(\boldsymbol{Y}'\boldsymbol{Q}^* - \boldsymbol{Y}) \in \mathbb{R}^{PR}$, where the vect of a matrix is the vector containing all its coefficients.

This framework offers closed form projections in the tangent space for the Wasserstein distance, which can be used to perform regression. Importantly, since $S_P^{++} = S_{P,P}^+$, we can also use this distance on the positive definite matrices. This distance possesses the *orthogonal invariance* property:

$$\text{For } \boldsymbol{W} \text{ orthogonal}, d_W(\boldsymbol{W}^\top \boldsymbol{SW}, \boldsymbol{W}^\top \boldsymbol{S}'\boldsymbol{W}) = d_W(\boldsymbol{S}, \boldsymbol{S}') \ . \tag{10}$$

This property is weaker than the affine invariance of the geometric distance (7). A natural question is whether such an affine invariant distance also exists on this manifold. Unfortunately, it is shown in [8] that the answer is negative for $R < P$ (proof in appendix 6.3).

## 3  Manifold-regression models for M/EEG

### 3.1  Generative model and consistency of linear regression in the tangent space of $\mathcal{S}_P^{++}$

Here, we consider a more specific generative model than (1) by assuming a specific structure on the noise. We assume that the additive noise $\boldsymbol{n}_i(t) = \boldsymbol{A}^n \boldsymbol{\nu}_i(t)$ with $\boldsymbol{A}^n = [\boldsymbol{a}_1^n, \ldots, \boldsymbol{a}_{P-Q}^n] \in \mathbb{R}^{P \times (P-Q)}$ and $\boldsymbol{\nu}_i(t) \in \mathbb{R}^{P-Q}$. This amounts to assuming that the noise is of rank $P - Q$ and that the noise spans the same subspace for all subjects. Denoting $\boldsymbol{A} = [\boldsymbol{a}_1^s, \ldots, \boldsymbol{a}_Q^s, \boldsymbol{a}_1^n, \ldots, \boldsymbol{a}_{P-Q}^n] \in \mathbb{R}^{P \times P}$ and $\boldsymbol{\eta}_i(t) = [s_{i,1}(t), \ldots s_{i,Q}(t), \nu_{i,1}(t), \ldots, \nu_{i,P-Q}(t)] \in \mathbb{R}^P$, this generative model can be compactly rewritten as $\boldsymbol{x}_i(t) = \boldsymbol{A}\boldsymbol{\eta}_i(t)$.

We assume that the sources $\boldsymbol{s}_i$ are decorrelated and independent from $\boldsymbol{\nu}_i$: with $p_{i,j} = \mathbb{E}_t[s_{i,j}^2(t)]$ the powers, *i.e.* the variance over time, of the $j$-th source of subject $i$, we suppose $\mathbb{E}_t[\boldsymbol{s}_i(t)\boldsymbol{s}_i^\top(t)] = \text{diag}((p_{i,j})_{j=1\ldots Q})$ and $\mathbb{E}_t[\boldsymbol{s}_i(t)\boldsymbol{\nu}_i(t)^\top] = 0$. The covariances are then given by:

$$\boldsymbol{C}_i = \boldsymbol{A}\boldsymbol{E}_i\boldsymbol{A}^\top \ , \tag{11}$$

where $\boldsymbol{E}_i = \mathbb{E}_t[\boldsymbol{\eta}_i(t)\boldsymbol{\eta}_i(t)^\top]$ is a block diagonal matrix, whose upper $Q \times Q$ block is $\text{diag}(p_{i,1}, \ldots, p_{i,Q})$.

In the following, we show that different functions $f$ from (2) yield a linear relationship between the $y_i$'s and the vectorization of the $\boldsymbol{C}_i$'s for different Riemannian metrics.

**Proposition 1** (Euclidean vectorization). *Assume $f(p_{i,j}) = p_{i,j}$. Then, the relationship between $y_i$ and $Upper(\boldsymbol{C}_i)$ is linear.*

*Proof.* Indeed, if $f(p) = p$, the relationship between $y_i$ and the $p_{i,j}$ is linear. Rewriting Eq. (11) as $\boldsymbol{E}_i = \boldsymbol{A}^{-1}\boldsymbol{C}_i\boldsymbol{A}^{-\top}$, and since the $p_{i,j}$ are on the diagonal of the upper block of $\boldsymbol{E}_i$, the relationship between the $p_{i,j}$ and the coefficients of $\boldsymbol{C}_i$ is also linear. This means that there is a linear relationship between the coefficients of $\boldsymbol{C}_i$ and the variable of interest $y_i$. In other words, $y_i$ is a linear combination of the vectorization of $\boldsymbol{C}_i$ *w.r.t.* the standard Euclidean distance. □

**Proposition 2** (Geometric vectorization). *Assume $f(p_{i,j}) = \log(p_{i,j})$. Denote $\overline{\boldsymbol{C}} = Mean_G(\boldsymbol{C}_1, \ldots, \boldsymbol{C}_N)$ the geometric mean of the dataset, and $\boldsymbol{v}_i = \mathcal{P}_{\overline{\boldsymbol{C}}}(\boldsymbol{C}_i)$ the vectorization of $\boldsymbol{C}_i$* w.r.t. *the geometric distance. Then, the relationship between $y_i$ and $\boldsymbol{v}_i$ is linear.*

The proof is given in appendix 6.1. It relies crucially on the affine invariance property that means that using Riemannian embeddings of the $\boldsymbol{C}_i$'s, is equivalent to working directly with the $\boldsymbol{E}_i$'s.

**Proposition 3** (Wasserstein vectorization). *Assume $f(p_{i,j}) = \sqrt{p_{i,j}}$. Assume that $\boldsymbol{A}$ is orthogonal. Denote $\overline{\boldsymbol{C}} = Mean_W(\boldsymbol{C}_1, \ldots, \boldsymbol{C}_N)$ the Wasserstein mean of the dataset, and $\boldsymbol{v}_i = \mathcal{P}_{\overline{\boldsymbol{C}}}(\boldsymbol{C}_i)$ the vectorization of $\boldsymbol{C}_i$* w.r.t. *the Wasserstein distance. Then, the relationship between $y_i$ and $\boldsymbol{v}_i$ is linear.*

The proof is given in appendix 6.2. The restriction to the case where $A$ is orthogonal stems from the orthogonal invariance of the Wasserstein distance. In the neuroscience literature square root rectifications are however not commonly used for M/EEG modeling. Nevertheless, it is interesting to see that the Wasserstein metric that can naturally cope with rank reduced data is consistent with this particular generative model.

These propositions show that the relationship between the samples and the variable $y$ is linear in the tangent space, motivating the use of linear regression methods (see simulation study in Sec. 4). The argumentation of this section relies on the assumption that the covariance matrices are full rank. However, this is rarely the case in practice.

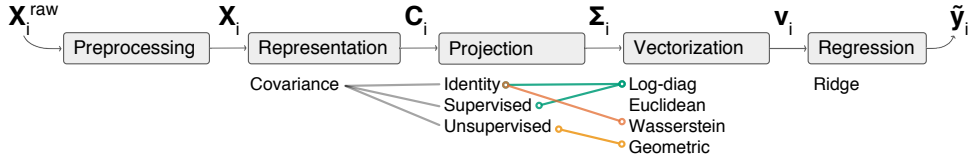

Figure 2: Proposed regression pipeline. The considered choices for each sequential step are detailed below each box. Identity means no spatial filtering $\boldsymbol{W} = \boldsymbol{I}$. Only the most relevant combinations are reported. For example Wasserstein vectorization does not need projections as it directly applies to rank-deficient matrices. Geometric vectorization is not influenced by the choice of projections due to its affine-invariance. Choices for vectorization are depicted by the colors used for visualizing subsequent analyses.

## 3.2 Learning projections on $\mathcal{S}_R^{++}$

In order to use the geometric distance on the $\boldsymbol{C}_i \in \mathcal{S}_{P,R}^+$, we have to project them on $\mathcal{S}_R^{++}$ to make them full rank. In the following, we consider a linear operator $\boldsymbol{W} \in \mathbb{R}^{P \times R}$ of rank $R$ which is common to all samples (*i.e.* subjects). For consistency with the M/EEG literature we will refer to rows of $\boldsymbol{W}$ as *spatial filters*. The covariance matrices of 'spatially filtered' signals $\boldsymbol{W}^\top \boldsymbol{x}_i$ are obtained as: $\boldsymbol{\Sigma}_i = \boldsymbol{W}^\top \boldsymbol{C}_i \boldsymbol{W} \in \mathbb{R}^{R \times R}$. With probability one, $\mathrm{rank}(\boldsymbol{\Sigma}_i) = \min(\mathrm{rank}(\boldsymbol{W}), \mathrm{rank}(\boldsymbol{C}_i)) = R$, hence $\boldsymbol{\Sigma}_i \in S_R^{++}$. Since the $\boldsymbol{C}_i$'s do not span the same image, applying $\boldsymbol{W}$ destroys some information. Recently, geometry-aware dimensionality reduction techniques, both supervised and unsupervised, have been developed on covariance manifolds [28, 25]. Here we considered two distinct approaches to estimate $\boldsymbol{W}$.

**Unsupervised spatial filtering** A first strategy is to project the data into a subspace that captures most of its variance. This is achieved by Principal Component Analysis (PCA) applied to the averaged covariance matrix computed across subjects: $\boldsymbol{W}_{\mathrm{UNSUP}} = \boldsymbol{U}$, where $\boldsymbol{U}$ contains the eigenvectors corresponding to the top $R$ eigenvalues of the average covariance matrix $\overline{\boldsymbol{C}} = \frac{1}{N} \sum_{i=1}^N \boldsymbol{C}_i$. This step is blind to the values of $y$ and is therefore unsupervised. Note that under the assumption that the time series across subjects are independent, the average covariance $\overline{\boldsymbol{C}}$ is the covariance of the data over the full population.

**Supervised spatial filtering** We use a supervised spatial filtering algorithm [15] originally developed for intra-subject Brain Computer Interfaces applications, and adapt it to our cross-person prediction problem. The filters $\boldsymbol{W}$ are chosen to maximize the covariance between the power of the filtered signals and $y$. Denoting by $\boldsymbol{C}_y = \frac{1}{N} \sum_{i=1}^N y_i \boldsymbol{C}_i$ the weighted average covariance matrix, the first filter $\boldsymbol{w}_{\mathrm{SUP}}$ is given by:

$$\boldsymbol{w}_{\mathrm{SUP}} = \arg\max_{\boldsymbol{w}} \frac{\boldsymbol{w}^\top \boldsymbol{C}_y \boldsymbol{w}}{\boldsymbol{w}^\top \overline{\boldsymbol{C}} \boldsymbol{w}} \ .$$

In practice, all the other filters in $\boldsymbol{W}_{\mathrm{SUP}}$ are obtained by solving a generalized eigenvalue decomposition problem (see the proof in Appendix 6.4).

The proposed pipeline is summarized in Fig. 2.

## 4 Experiments

### 4.1 Simulations

We start by illustrating Prop. 2. Independent identically distributed covariance matrices $\boldsymbol{C}_1, \ldots, \boldsymbol{C}_N \in S_P^{++}$ and variables $y_1, \ldots, y_N$ are generated following the above generative model. The matrix $\boldsymbol{A}$ is taken as $\exp(\mu \boldsymbol{B})$ with $\boldsymbol{B} \in \mathbb{R}^{P \times P}$ a random matrix, and $\mu \in \mathbb{R}$ a scalar controlling the distance from $A$ to identity ($\mu = 0$ yields $\boldsymbol{A} = \boldsymbol{I}_P$). We use the $\log$ function for $f$ to link the source powers (*i.e.* the variance) to the $y_i$'s. Model reads $y_i = \sum_j \alpha_j \log(p_{ij}) + \varepsilon_i$, with $\varepsilon_i \sim \mathcal{N}(0, \sigma^2)$ a small additive random perturbation.

We compare three methods of vectorization: the geometric distance, the Wasserstein distance and the non-Riemannian method "log-diag" extracting the $\log$ of the diagonals of $C_i$ as features. Note that the diagonal of $C_i$ contains the powers of each sensor for subject $i$. A linear regression model is used following the procedure presented in Sec. 2. We take $P = 5$, $N = 100$ and $Q = 2$. We measure the score of each method as the average mean absolute error (MAE) obtained with 10-fold cross-validation. Fig. 3 displays the scores of each method when the parameters $\sigma$ controlling the noise level and $\mu$ controlling the distance from $A$ to $I_p$ are changed. We also investigated the realistic scenario where each subject has a mixing matrix deviating from a reference: $\boldsymbol{A}_i = \boldsymbol{A} + \boldsymbol{E}_i$ with entries of $\boldsymbol{E}_i$ sampled i.i.d. from $\mathcal{N}(0, \sigma^2)$.

The same experiment with $f(p) = \sqrt{p}$ yields comparable results, yet with Wasserstein distance performing best and achieving perfect out-of-sample prediction when $\sigma \to 0$ and $A$ is orthogonal.

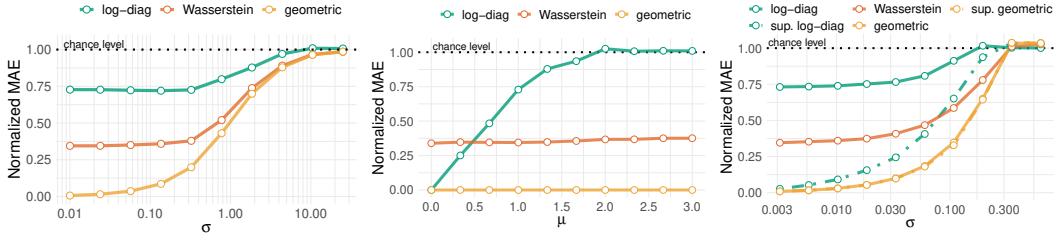

Figure 3: Illustration of Prop.2. Data is generated following the generative model with $f = \log$. The regression pipeline consists in projecting the data in the tangent space, and then use a linear model. The left plot shows the evolution of the score when random noise of variance $\sigma^2$ is added to the variables $y_i$. The MAE of the geometric distance pipeline goes to 0 in the limit of no noise, indicating perfect out-of-sample prediction. This illustrates the linearity in the tangent space for the geometric distance (Prop. 2). The middle plot explores the effect of the parameter $\mu$ controlling the distance between $A$ and $I_P$. Riemannian geometric method is not affected by $\mu$ due to its affine invariance property. Although the Wasserstein distance is not affine invariant, its performance does not change much with $\mu$. On the contrary, the log-diag method is sensitive to changes in $A$. The right plot shows how the score changes when mixing matrices become sample dependent. We can see then only when $\sigma = 0$ supervised + log-diag and Riemann reach perfect performance. Geometric Riemann is uniformly better and indifferent to projection choice. Wasserstein, despite model mismatch, outperforms supervised + log-diag with high $\sigma$.

## 4.2 MEG data

**Predicting biological age from MEG on the Cambridge center of ageing dataset** In the following, we apply our methods to infer age from brain signals. Age is a dominant driver of cross-person variance in neuroscience data and a serious confounder [39]. As a consequence of the globally increased average lifespan, ageing has become a central topic in public health that has stimulated neuropsychiatric research at large scales. The link between age and brain function is therefore of utmost practical interest in neuroscientific research.

To predict age from brain signals, here we use the currently largest publicly available MEG dataset provided by the Cam-CAN [38]. We only considered the signals from magnetometer sensors ($P = 102$) as it turns out that once SSS is applied (detailed in Appendix 6.6), magnetometers and gradiometers are linear combination of approximately 70 signals ($65 \leq R_i \leq 73$), which become redundant in practice [19]. We considered task-free recordings during which participants were asked to sit still with eyes closed in the absence of systematic stimulation. We then drew $T \simeq 520,000$ time samples from $N = 595$ subjects. To capture age-related changes in cortical brain rhythms [4, 44, 12], we filtered the data into 9 frequency bands: low frequencies $[0.1-1.5]$, $\delta[1.5-4]$, $\theta[4-8]$, $\alpha[8-15]$, $\beta_{low}[15-26]$, $\beta_{high}[26-35]$, $\gamma_{low}[35-50]$, $\gamma_{mid}[50-74]$ and $\gamma_{high}[76-120]$ (Hz unit). These frequencies are compatible with conventional definitions used in the Human Connectome Project [32]. We verify that the covariance matrices all lie on a small portion of the manifold, justifying projection in a common tangent space. Then we applied the covariance pipeline independently in each frequency band and concatenated the ensuing features.

**Data-driven covariance projection for age prediction**    Three types of approaches are here compared: Riemannian methods (Wasserstein or geometric), methods extracting log-diagonal of matrices (with or without supervised spatial filtering, see Sec. 3.2) and a biophysics-informed method based on the MNE source imaging technique [24]. The MNE method essentially consists in a standard Tikhonov regularized inverse solution and is therefore linear (See Appendix 6.5 for details). Here it serves as gold-standard informed by the individual anatomy of each subject. It requires a T1-weighted MRI and the precise measure of the head in the MEG device coordinate system [3] and the coordinate alignment is hard to automate. We configured MNE with $Q = 8196$ candidate dipoles. To obtain spatial smoothing and reduce dimensionality, we averaged the MNE solution using a cortical parcellation encompassing 448 regions of interest from [31, 21]. We then used ridge regression and tuned its regularization parameter by generalized cross-validation [20] on a logarithmic grid of 100 values in $[10^{-5}, 10^3]$ on each training fold of a 10-fold cross-validation loop. All numerical experiments were run using the Scikit-Learn software [36], the MNE software for processing M/EEG data [21] and the PyRiemann package [13]. We also ported to Python some part of the Matlab code of Manopt toolbox [9] for computations involving Wasserstein distance. The proposed method, including all data preprocessing, applied on the 500GB of raw MEG data from the Cam-CAN dataset, runs in approximately 12 hours on a regular desktop computer with at least 16GB of RAM. The preprocessing for the computation of the covariances is embarrassingly parallel and can therefore be significantly accelerated by using multiple CPUs. The actual predictive modeling can be performed in less than a minute on standard laptop. Code used for data analysis can be found on GitHub[5].

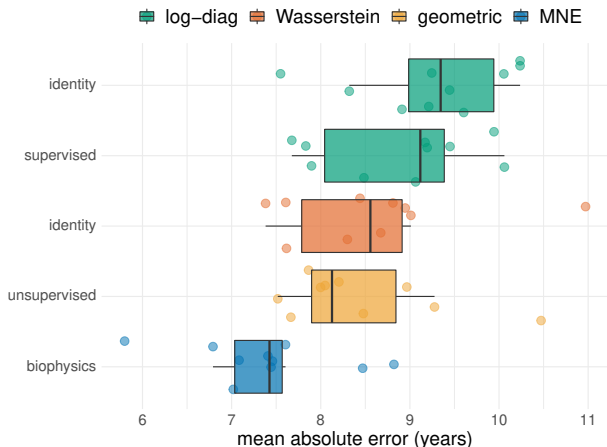

Figure 4: Age prediction on Cam-CAN MEG dataset for different methods, ordered by out-of-sample MAE. The y-axis depicts the projection method, with identity denoting the absence of projection. Colors indicate the subsequent embedding. The biophysics-driven MNE method (blue) performs best. The Riemannian methods (orange) follow closely and their performance depends little on the projection method. The non-Riemannian methods log-diag (green) perform worse, although the supervised projection clearly helps.

**Riemannian projections are the leading data-driven methods**    Fig. 4 displays the scores for each method. The biophysically motivated MNE projection yielded the best performance (7.4y MAE), closely followed by the purely data-driven Riemannian methods (8.1y MAE). The chance level was 16y MAE. Interestingly, the Riemannian methods give similar results, and outperformed the non-Riemannian methods. When Riemannian geometry was not applied, the projection strategy turned out to be decisive. Here, the supervised method performed best: it reduced the dimension of the problem while preserving the age-related variance.

Rejecting a null-hypothesis that differences between models are due to chance would require several independent datasets. Instead, for statistical inference, we considered uncertainty estimates of paired differences using 100 Monte Carlo splits (10% test set size). For each method, we counted how often it was performing better than the baseline model obtained with identity and log-diag. We observed for supervised log-diag 73%, identity Wasserstein 85%, unsupervised geometric 96% and biophysics 95% improvement over baseline. This suggests that inferences will carry over to new data.

Importantly, the supervised spatial filters and MNE both support model inspection, which is not the case for the two Riemannian methods. Fig. 5 depicts the marginal patterns [27] from the supervised filters and the source-level ridge model, respectively. The sensor-level results suggest predictive dipolar patterns in the theta to beta range roughly compatible with generators in visual, auditory and motor cortices. Note that differences in head-position can make the sources appear deeper than

[5] https://www.github.com/DavidSabbagh/NeurIPS19_manifold-regression-meeg

they are (distance between the red positive and the blue negative poles). Similarly, the MNE-based model suggests localized predictive differences between frequency bands highlighting auditory, visual and premotor cortices. While the MNE model supports more exhaustive inspection, the supervised patterns are still physiologically informative. For example, one can notice that the pattern is more anterior in the $\beta$-band than the $\alpha$-band, potentially revealing sources in the motor cortex.

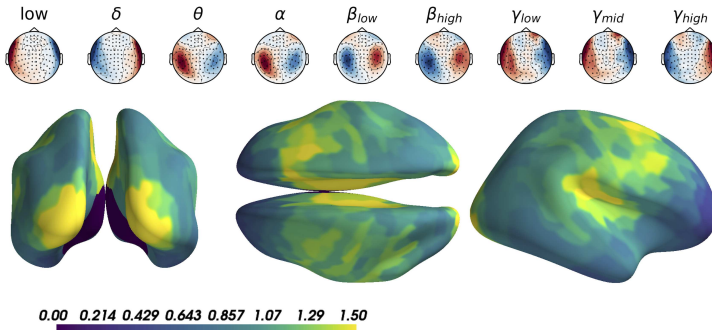

Figure 5: Model inspection. Upper panel: sensor-level patterns from supervised projection. One can notice dipolar configurations varying across frequencies. Lower panel: standard deviation of patterns over frequencies from MNE projection highlighting bilateral visual, auditory and premotor cortices.

## 5 Discussion

In this contribution, we proposed a mathematically principled approach for regression on rank-reduced covariance matrices from M/EEG data. We applied this framework to the problem of inferring age from neuroimaging data, for which we made use of the currently largest publicly available MEG dataset. To the best of our knowledge, this is the first study to apply a covariance-based approach coupled with Riemannian geometry to regression problem in which the target is defined across persons and not within persons (as in brain-computer interfaces). Moreover, this study reports the first benchmark of age prediction from MEG resting state data on the Cam-CAN. Our results demonstrate that Riemannian data-driven methods do not fall far behind the gold-standard methods with biophysical priors, that depend on manual data processing. One limitation of Riemannian methods is, however, their interpretability compared to other models that allow to report brain-region and frequency-specific effects. These results suggest a trade-off between performance and explainability. Our study suggests that the Riemannian methods have the potential to support automated large-scale analysis of M/EEG data in the absence of MRI scans. Taken together, this potentially opens new avenues for biomarker development.

## Acknowledgement

This work was supported by a 2018 *"médecine numérique" (for digital Medicine)* thesis grant issued by Inserm (French national institute of health and medical research) and Inria (French national research institute for the digital sciences). It was also partly supported by the European Research Council Starting Grant SLAB ERC-YStG-676943.

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
