[Supplementary Material]

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

, which depend on manual data processing. Finally, we report models that are explainable as they allow to uncover brain-region and frequency-band specific effects. These results suggest a trade-off between performance and explainability. Our study suggests that the Riemannian methods have the potential to support automated large-scale analysis of M/EEG data in the absence of MRI scans. Taken together, this potentially opens new avenues for biomarker development.

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

# 6  Appendix

## 6.1  Proof of proposition 2

First, we note that by invariance, $\overline{C} = \mathrm{Mean}_G(C_1, \ldots, C_N) = A\mathrm{Mean}_G(E_1, \ldots, E_N)A^\top = A\overline{E}A^\top$, where $\overline{E}$ has the same block diagonal structure as the $E_i$'s, and $\overline{E}_{jj} = (\prod_{i=1}^N p_{i,j})^{\frac{1}{N}}$ for $j \leq Q$. Denote $\overline{U} = \overline{C}^{\frac{1}{2}}A^{-\top}\overline{E}^{-\frac{1}{2}}$. By simple verification, we obtain $\overline{U}^\top \overline{U} = I_P$, i.e. $\overline{U}$ is orthogonal.

Furthermore, we have:
$$\overline{U}^\top \overline{C}^{-\frac{1}{2}} C_i \overline{C}^{-\frac{1}{2}} \overline{U} = \overline{E}^{-\frac{1}{2}} E_i \overline{E}^{-\frac{1}{2}} \ .$$

It follows that for all $i$,
$$\overline{U}^\top \log(\overline{C}^{-\frac{1}{2}} C_i \overline{C}^{-\frac{1}{2}})\overline{U} = \log(\overline{E}^{-\frac{1}{2}} E_i \overline{E}^{-\frac{1}{2}})$$

Note that $\log(\overline{E}^{-\frac{1}{2}} E_i \overline{E}^{-\frac{1}{2}})$ shares the same structure as the $E_i$'s, and that $\log(\overline{E}^{-\frac{1}{2}} E_i \overline{E}^{-\frac{1}{2}})_{jj} = \log(\frac{p_{i,j}}{\overline{p}_j})$. for $j \leq Q$.

Therefore, the relationship between $\log(\overline{C}^{-\frac{1}{2}} C_i \overline{C}^{-\frac{1}{2}})$ and the $\log(p_{i,j})$ is linear.

Finally, since $v_i = \mathrm{Upper}(\log(\overline{C}^{-\frac{1}{2}} C_i \overline{C}^{-\frac{1}{2}}))$, the relationship between the $v_i$'s and the $\log(p_{i,j})$ is linear, and the result holds.

## 6.2  Proof of proposition 3

First, we note that $C_i = AE_iA^\top \in S_P^{++} = S_{P,P}^+$ so it can be decomposed as $C_i = Y_iY_i^\top$ with $Y_i = AE_i^{\frac{1}{2}}$.

By orthogonal invariance, $\overline{C} = \mathrm{Mean}_W(C_1, \ldots, C_N) = A\mathrm{Mean}_W(E_1, \ldots, E_N)A^\top = A\overline{E}A^\top$, where $\overline{E}$ so has the same block diagonal structure as the $E_i$'s, and $\overline{E}_{jj} = (\sum_i \sqrt{p_{ij}})^2$ for $j \leq Q$. $\overline{C}$ is also decomposed as $\overline{C} = \overline{Y}\,\overline{Y}^\top$ with $\overline{Y} = A\overline{E}^{\frac{1}{2}}$.

Further, $Q_i^* = V_iU_i^\top$ with $U_i$ and $V_i$ coming from the SVD of $\overline{Y}^\top Y_i = \overline{E}^{\frac{1}{2}} E_i^{\frac{1}{2}}$ which has the same structure as the $E_i$'s. Therefore $Q_i^*$ has also the same structure with the identity matrix as its upper block.

Finally we have $v_i = \mathcal{P}_{\overline{C}}(C_i) = \mathrm{vect}(Y_iQ_i^* - \overline{Y})$ so it is linear in $\sqrt{(p_{i,j})}$ for $j \leq Q$.

## 6.3  Proof that there is no continuous affine invariant distance on $S_{P,R}^+$ if $R < P$

We show the result for $P = 2$ and $R = 1$; the demonstration can straightforwardly be extended to the other cases. The proof, from [8], is by contradiction.

Assume that $d$ is a continuous invariant distance on $S_{2,1}^+$. Consider $A = \begin{pmatrix} 1 & 0 \\ 0 & 0 \end{pmatrix}$ and $B = \begin{pmatrix} 1 & 1 \\ 1 & 1 \end{pmatrix}$, both in $S_{2,1}^+$. For $\varepsilon > 0$, consider the invertible matrix $W_\varepsilon = \begin{pmatrix} 1 & 0 \\ 0 & \varepsilon \end{pmatrix}$.

We have: $W_\varepsilon A W_\varepsilon^\top = A$, and $W_\varepsilon B W_\varepsilon^\top = \begin{pmatrix} 1 & \varepsilon \\ \varepsilon & \varepsilon^2 \end{pmatrix}$.

Hence, as $\varepsilon$ goes to $0$, we have $W_\varepsilon B W_\varepsilon^\top \to A$

Using affine invariance, we have:
$$d(A, B) = d(W_\varepsilon A W_\varepsilon^\top, W_\varepsilon B W_\varepsilon^\top)$$

Letting $\varepsilon \to 0$ and using continuity of $d$ yields $d(A, B) = d(A, A) = 0$, which is absurd since $A \neq B$.

## 6.4 Supervised Spatial Filtering

465 We assume that the signal $\boldsymbol{x}(t)$ is band-pass filtered in one of frequency band of interest, so that for
466 each subject the band power of signal is approximated by the variance over time of the signal. We
467 denote the expectation $\mathbb{E}$ and the variance $\mathbb{V}\mathrm{ar}$ over time $t$ or subject $i$ by a corresponding subscript.

468 The source extracted by a spatial filter $\boldsymbol{w}$ for subject $i$ is $\widehat{s}_i = \boldsymbol{w}^\top \boldsymbol{x}_i(t)$. Its power reads:

$$\Phi_i^{\boldsymbol{w}} = \mathbb{V}\mathrm{ar}_t[\boldsymbol{w}^\top \boldsymbol{x}_i(t)] = \mathbb{E}_t[\boldsymbol{w}^\top \boldsymbol{x}_i(t)\boldsymbol{x}_i^\top(t)\boldsymbol{w}] = \boldsymbol{w}^\top \boldsymbol{C}_i \boldsymbol{w}$$

469 and its expectation across subjects is given by:

$$\mathbb{E}_i[\Phi_i^{\boldsymbol{w}}] = \boldsymbol{w}^\top \mathbb{E}_i[\boldsymbol{C}_i]\boldsymbol{w} = \boldsymbol{w}^\top \overline{\boldsymbol{C}} \boldsymbol{w} \ ,$$

470 where $\overline{\boldsymbol{C}} = \frac{1}{N}\sum_i \boldsymbol{C}_i$ is the average covariance matrix across subjects. Note that here, $\boldsymbol{C}_i$ refers to
471 the covariance of the $\boldsymbol{x}_i$ and not its estimate as in Sec. 3.2.

We aim to maximize the covariance between the target $y$ and the power of the sources, $\mathbb{C}\mathrm{ov}_i[\Phi_i^{\boldsymbol{w}}, y_i]$.
This quantity is affected by the scaling of its arguments. To address this, the target variable $y$ is
normalized:

$$\mathbb{E}_i[y_i] = 0 \quad \mathbb{V}\mathrm{ar}_i[y_i] = 1 \ .$$

Following [13], to also scale $\Phi_i^{\boldsymbol{w}}$ we constrain its expectation to be 1:

$$\mathbb{E}_i[\Phi_i^{\boldsymbol{w}}] = \boldsymbol{w}^\top \overline{\boldsymbol{C}} \boldsymbol{w} = 1$$

472 The quantity one aims to maximize reads:

$$\begin{aligned}
\mathbb{C}\mathrm{ov}_i[\Phi_i^{\boldsymbol{w}}, y_i] &= \mathbb{E}_i[\ (\Phi_i^{\boldsymbol{w}} - \mathbb{E}_i[\Phi_i^{\boldsymbol{w}}])\ (y_i - \mathbb{E}_i[y_i])\ ] \\
&= \boldsymbol{w}^\top \mathbb{E}_i[\boldsymbol{C}_i y_i]\boldsymbol{w} - \boldsymbol{w}^\top \overline{\boldsymbol{C}} \boldsymbol{w}\mathbb{E}_i[y_i] \\
&= \boldsymbol{w}^\top \boldsymbol{C}_y \boldsymbol{w}
\end{aligned}$$

473 where $\boldsymbol{C}_y = \frac{1}{N}\sum_i y_i \boldsymbol{C}_i$.

474

475 Taking into account the normalization constraint we obtain:

$$\widehat{\boldsymbol{w}} = \underset{\boldsymbol{w}^\top \overline{\boldsymbol{C}}\boldsymbol{w}=1}{\arg\max}\ \boldsymbol{w}^\top \boldsymbol{C}_y \boldsymbol{w} \ . \tag{12}$$

476 The Lagrangian of (12) reads $F(\boldsymbol{w}, \lambda) = \boldsymbol{w}^\top \boldsymbol{C}_y \boldsymbol{w} + \lambda(1 - \boldsymbol{w}^\top \overline{\boldsymbol{C}} \boldsymbol{w})$. Setting its gradient *w.r.t.* $\boldsymbol{w}$
477 to 0 yields a generalized eigenvalue problem:

$$\nabla_{\boldsymbol{w}} F(\boldsymbol{w}, \lambda) = 0 \implies \Sigma_y \boldsymbol{w} = \lambda \overline{\Sigma_{\boldsymbol{x}}} \boldsymbol{w} \tag{13}$$

478 Note that (12) can be also written as a generalized Rayleigh quotient:

$$\widehat{\boldsymbol{w}} = \underset{\boldsymbol{w}}{\arg\max}\ \frac{\boldsymbol{w}^\top \boldsymbol{C}_y \boldsymbol{w}}{\boldsymbol{w}^\top \overline{\boldsymbol{C}} \boldsymbol{w}} \ .$$

479 Equation (13) has a unique closed-form solution called the generalized eigenvectors of $(\boldsymbol{C}_y, \overline{\boldsymbol{C}})$. The
480 second derivative gives:

$$\nabla_{\boldsymbol{\lambda}} F(\boldsymbol{w}, \lambda) = 0 \implies \lambda = \boldsymbol{w}^\top \Sigma_y \boldsymbol{w} = \mathbb{C}\mathrm{ov}_i[\Phi_i^{\boldsymbol{w}}, y_i] \tag{14}$$

481 Equation (14) leads to an interpretation of $\lambda$ as the covariance between $\Phi^{\boldsymbol{w}}$ and $y$, which should be
482 maximal. As a consequence, $\boldsymbol{W}_{\mathrm{SUP}}$ is built from the generalized eigenvectors of Eq.(13), sorted by
483 decreasing eigenvalues.

## 6.5 MNE-based spatial filtering

Let us denote $\boldsymbol{G} \in \mathbb{R}^{P \times Q}$ the instantaneous mixing matrix that relates the sources in the brain to
the MEG/EEG measurements. This forward operator matrix is obtained by solving numerically
Maxwell's equations after specifying a geometrical model of the head, typically obtained using an
anatomical MRI image [22]. Here $Q \geq P$ corresponds to the number of candidate sources in the
brain. The MNE approach [21] offers a way to solve the inverse problem. MNE can be seen as
Tikhonov regularized estimation, also similar to a ridge regression in statistics. Using such problem
formulation the sources are obtained from the measurements with a linear operator which is given by:

$$\boldsymbol{W}_{\mathrm{MNE}} = \boldsymbol{G}^\top (\boldsymbol{G}\boldsymbol{G}^\top + \lambda \boldsymbol{I}_P)^{-1} \in \mathbb{R}^{Q \times P} \ .$$

485 The rows of this linear operator $\boldsymbol{W}_{\mathrm{MNE}}$ can be seen also as spatial filters that are mapped to specific
486 locations in the brain. These are the filters used in Fig. 3, using the implementation from [18].

## 6.6 Preprocessing

Typical brain's magnetic fields detected by MEG are in the order of 100 femtotesla ($1fT = 10^{-15}$ T) which is ~$10^{-8}$ times the strength of the earth's steady magnetic field. That is why MEG recordings are carried out inside special magnetically shielded rooms (MSR) that eliminate or at least dampen external ambient magnetic disturbances.

To pick up such tiny magnetic fields sensitive sensors have to be used [22]. Their extreme sensitivity is challenged by many electromagnetic nuisance sources (any moving metal objects like cars or elevators) or electrically powered instruments generating magnetic induction that is orders of magnitude stronger than the brain's. Their influence can be reduced by combining magnetometers coils (that directly record the magnetic field) with gradiometers coils (that record the gradient of the magnetic field in certain directions). Those gradiometers, arranged either in a radial or tangential (planar) way, record the gradient of the magnetic field towards 2 perpendicular directions hence inherently greatly emphasize brain signals with respect to environmental noise.

Even though the magnetic shielded room and gradiometer coils can help to reduce the effects of external interference signals the problem mainly remains and further reduction is needed. Also additional artifact signals can be caused by movement of the subject during recording if the subject has small magnetic particles on his body or head. The Signal Space Separation (SSS) method can help mitigate those problems [34].

**Signal Space Separation (SSS)**  The Signal Space Separation (SSS) method [34], also called Maxwell Filtering, is a biophysical spatial filtering method that aim to produce signals cleaned from external interference signals and from movement distortions and artifacts.

A MEG device records the neuromagnetic field distribution by sampling the field simultaneously at P distinct locations around the subject's head. At each moment of time the measurement is a vector $\boldsymbol{x} \in \mathbb{R}^P$ is the total number of recording channels.

In theory, any direction of this vector in the signal space represents a valid measurement of a magnetic field, however the knowledge of the location of possible sources of magnetic field, the geometry of the sensor array and electromagnetic theory (Maxwell's equations and the quasistatic approximation) considerably constrain the relevant signal space and allow us to differentiate between signal space directions consistent with a brain's field and those that are not.

To be more precise, it has been shown that the recorded magnetic field is a gradient of a harmonic scalar potential. A harmonic potential $V(\boldsymbol{r})$ is a solution of the Laplacian differential equation $\nabla^2 V = 0$, where $\boldsymbol{r}$ is represented by its spherical coordinates $(r, \theta, \psi)$. It has been shown that any harmonic function in a three-dimensional space can be represented as a series expansion of spherical harmonic functions $Y_{lm}(\theta, \phi)$:

$$V(\boldsymbol{r}) = \sum_{l=1}^{\infty} \sum_{m=-l}^{l} \alpha_{lm} \frac{Y_{lm}(\theta, \phi)}{r^{l+1}} + \sum_{l=1}^{\infty} \sum_{m=-l}^{l} \beta_{lm} r^l Y_{lm}(\theta, \phi) \tag{15}$$

We can separate this expansion into two sets of functions: those proportional to inverse powers of $r$ and those proportional to powers of $r$. From a given array of sensors and a coordinate system with its origin somewhere inside of the helmet, we can compute the signal vectors corresponding to each of the terms in 15.

Following notations of [34], let $\boldsymbol{a}_{lm}$ be the signal vector corresponding to term $\frac{Y_{lm}(\theta, \phi)}{r^{l+1}}$ and $\boldsymbol{b}_{lm}$ the signal vector corresponding to $r^l Y_{lm}(\theta, \phi)$. A set of P such signal vectors forms a basis in the P dimensional signal space, and hence, the signal vector is given as

$$\boldsymbol{x} = \sum_{l=1}^{\infty} \sum_{m=-l}^{l} \alpha_{lm} \boldsymbol{a}_{lm} + \sum_{l=1}^{\infty} \sum_{m=-l}^{l} \beta_{lm} \boldsymbol{b}_{lm} \tag{16}$$

This basis is not orthogonal, but linearly independent so any measured signal vector has a unique representation in this basis:

$$\boldsymbol{x} = [\boldsymbol{S_{in}} \quad \boldsymbol{S_{out}}] \begin{bmatrix} \boldsymbol{x}_{in} \\ \boldsymbol{x}_{out} \end{bmatrix} \tag{17}$$

where the sub-bases $\boldsymbol{S}_{in}$ and $\boldsymbol{S}_{out}$ contain the basis vectors $\boldsymbol{a}_{lm}$ and $\boldsymbol{b}_{lm}$, and vectors $\boldsymbol{x}_{in}$ and $\boldsymbol{x}_{out}$ contain the corresponding $\alpha_{lm}$ and $\beta_{lm}$ values.

It can be shown that the spherical harmonic functions contain increasingly higher spatial frequencies when going to higher index values (l,m) so that the signals from real magnetic sources are mostly contained in the low $l, m$ end of the spectrum. By discarding the high $l, m$ end of the spectrum we thus reduce the noise. Then we can do signal space separation. It can be shown that the basis vectors corresponding to the terms in the second sum in expansion (15) represent the perturbating sources external to the helmet. We can than separate the components of field arising from sources inside and outside of the helmet. By discarding them we are left with the part of the signal coming from inside of the helmet only. The signal vector $\boldsymbol{x}$ is then decomposed into 2 components $\boldsymbol{\phi}_{in}$ and $\boldsymbol{\phi}_{out}$ with $\boldsymbol{\phi}_{in} = \boldsymbol{S}_{in}\boldsymbol{x}_{in}$ reproducing in all the MEG channels the signals that would be seen if no interference from sources external to the helmet existed.

The real data from the Cam-CAN dataset have been measured with an Elekta Neuromag 306-channel device, the only one that has been extensively tested on Maxwell Filtering. For this device we included components up to $l = L_{in} = 8$ for the $\boldsymbol{S}_{in}$ basis, and up to $l = L_{out} = 3$ for the $\boldsymbol{S}_{out}$ basis.

SSS requires a comprehensive sampling (more than about 150 channels) and a relatively high calibration accuracy that is machine/site-specific. For this purpose we used the fine-calibration coefficients and the cross-talk correction information provided in the Can-CAM repository for the 306-channels Neuromag system used in this study.

For this study we used the temporal SSS (tSSS) extension [34], where both temporal and spatial projection are applied to the MEG data. We used an order 8 (resp. 3) of internal (resp. external) component of spherical expansion, a 10s sliding window, a correlation threshold of 98% (limit between inner and outer subspaces used to reject overlapping intersecting inner/outer signals), basis regularization, no movement compensation.

The origin of internal and external multipolar moment space is fitted via head-digitization hence specified in the 'head' coordinate frame and the median head position during the 10s window is used.

After projection in the lower-dimensional SSS basis we project back the signal in its original space producing a signal $\boldsymbol{X}^{clean} = \boldsymbol{S}_{in}^{\top}\boldsymbol{S}_{in}\boldsymbol{X} \in \mathbb{R}^{P \times T}$ with a much better SNR (reduced noise variance) but with a rank $R \leq P$. As a result each reconstructed sensor is then a linear combination of $R$ synthetic source signals, which modifies the inter-channel correlation structure, rendering the covariance matrix rank-deficient.

**Signal Space Projection (SSP)**   Recalling the MEG generative model (1) if one knows, or can estimate, K linearly independent source patterns $\boldsymbol{a}_1, \ldots, \boldsymbol{a}_K$ that span the space $S = \mathrm{span}(\boldsymbol{a}_1, \ldots, \boldsymbol{a}_K) \subset \mathbb{R}^P$ that contains the brain signal, one can estimate an orthonormal basis $U_K \in \mathbb{R}^{P \times K}$ of $S$ by singular value decomposition (SVD). One can then project any sensor space signal $\boldsymbol{x} \in \mathbb{R}^P$ onto $S$ to improve the SNR. The projection reads:

$$U_K U_K^{\top} \boldsymbol{x} \ .$$

This is the idea behind the Signal Space Projections (SSP) method [36]. In practice SSP is used to reduce physiological artifacts (eye blinks and heart beats) that cause prominent artifacts in the recording. In the Cam-CAN dataset eye blinks are monitored by 2 electro-oculogram (EOG channels), and heart beats by an electro-cardiogram (ECG channel).

SSP projections are computed from time segments contaminated by the artifacts and the first component (per artifact and sensor type) are projected out. More precisely, the EOG and ECG channels are used to identify the artifact events (after a first band-pass filter to remove DC offset and an additional [1-10]Hz filter applied only to EOG channels to remove saccades vs blinks). After filtering the raw signal in [1-35]Hz band, data segments (called epochs) are created around those events, rejecting those whose peak-to-peak amplitude exceeds a certain global threshold (see section below). For each artifact and sensor type those epochs are then averaged and the first component of maximum variance is extracted via PCA. Signal is then projected in the orthogonal space. This follows the guidelines of the MNE software [18].

**Marking bad data segments**   We epoch the resulting data in 30s non overlapping windows and identify bad data segments (i.e. trials containing transient jumps in isolated channels) that have a

576 peak-to-peak amplitude exceeding a certain global threshold, learnt automatically from the data using
577 the autoreject (global) algorithm [24].