[Reviews · NeurIPS 2019]

Reviewer 1



Originality: although it is not groundbreaking, the paper is interesting to read. From the methodological point of view the originality comes from being the first paper showing how to perform regression with covariance matrices where the target is defined across persons. Information is missing on the difference from other techniques working with matrices for within-subject target and also comparison to Spoc is missing. Quality: the paper is quite easy to follow. I miss the following: 1) information on regression strategies within person and point out the differences to the method presented here. 2) a block diagram with the steps to be performed in each of the two cases they describe (for the covariance based approaches). 3) in case of rank deficient matrices: missing comparison between the method for rank deficient matrices versus methods for full rank matrices but using regularised covariance matrices. 4) significance results of the comparisons between methods. 5) comparison to SPoC. The paper states: "based on spoc method" but it is not clear if spoc could have been useful in this case nor how the methods would compare. 6) there is not clear claim about weaknesses of their approach. Clarity: the paper is clear, but some information is missing (see previous point). Significance: the paper is interesting for experts. It would fit the conference. It provides a new experimental approach. Clarifying its relation to Spoc might increase the significance of the paper.

Reviewer 2



The theoretical sections of the paper appear sound, with the Riemannian approaches and their respective invariance properties being properly established. The authors also discuss multiple possible functions that could be applied on the signal powers to obtain the target variable, and prove how using a linear regression model with the Riemannian feature vectors would be optimal for the identity, log and square roots of the signal power. However, they fail to discuss how often these types of scenarios occur in actual MEG/EEG dataset, and also how the performance would deteriorate in case where a different function of the source signals powers is used. The construction of the toy dataset is well thought out to exploit the invariances provided by the Riemannian metrics and demonstrate their performance in the ideal scenario. But as mentioned previously, some additional toy examples that examine the performance of the different models in sub-optimal conditions would also be useful. In addition, it would be interesting to see how the performance of the log-diag model on the toy dataset is affected by the use of supervised spacial filters, or how the geometric distance changes when supervised or unsupervised spacial filters are used. This comparison of the effects of supervised and unsupervised spacial filters with the geometric distance should also be performed on the Cam-CAN dataset evaluations, or the authors should at least address why this scenario was left out of the evaluation. Finally, the inclusion of the model inspection results seems unnecessary, since it does not relate to any of the methods proposed in this paper. Even though the Riemannian measures used in this paper are already existed, here they are applied in a new scenario, and good performance methods further demonstrates the usefulness of Riemannian techniques for MEG/EEG data analysis. The application of the geometric distance in conjunction with spatial filtering to deal with rank-reduced covariance matrices is an interesting approach, but it would be more interesting to see a bigger focus and a more detailed analysis on the interplay of the Riemannian metrics with the different spacial filters discussed in this paper. The claim that this paper presents the “first study to apply a covariance-based approach to regression problem in which the target is defined across persons and not within persons” is too broad and not entirely correct, as some other works can also be placed into this category (e.g. Dmochowski et al. 2012, Parra et al. 2019). However, the fact that this paper introduces the first benchmark on MEG-based age prediction for the Cam-CAN dataset could prove to be a significant contribution if this problem is seen as interesting by the community and is taken up as a standard benchmark for novel methods. The paper is generally clear and not too difficult to read, with the exception of sections 2.1 and 2.2 which are somewhat dense and more difficult to get through. Additionally, there are several small errors or typos throughout the text. No code was provided for the methods or evaluations, and while most of the details on the software used and preprocessing steps taken were discussed, but some specifics, such as if/how subjects were rejected and how the time-samples were drawn from the Cam-CAN dataset. A link to the toy dataset or corresponding code was also not provided.

Reviewer 3



The paper studies regression with rank-reduced covariance matrices from MEG data. Rank-reduced covariance matrices are an issue because they are the product of modern EEG/EMG artefact and noise reduction methods. The authors study two Riemannian approaches by projecting MEG covariance matrices into a tangent space: Firstly, the Wasserstein metric and secondly, a combination of linear projection and geometric distance. To evaluate the performance of the method the algorithm is applied to artificial data set and to the Cam-CAN dataset. Results of the proposed method are also compared to results of other state of the art methods. The results suggest that the proposed method is only outperformed by the complex biophysics driven source modelling. The proposed framework is novel and based on the Riemannian Geometry, which is becoming a more and more important approach used for the MEG/EEG analysis. Related work is properly cited. The paper is well structured and clear. The authors reasoning and rationale is sound. The discussion of pros and cons of the method could be increased. Particularly, because the used cross-validation approach may lead to an optimistic assessment of the regression. The reviewer understands that the whole Cam-CAN dataset was randomly sampled and that authors used a 10-fold cross-validation to test the regression. For example, a leave one-subject out cross-validation would have made a more realistic assessment of the performance. Nevertheless, the idea and the results are significant and other researchers will build on this idea. ======== Response to authors’ feedback: Thank you for the clarifications.

[Author Response · NeurIPS 2019]

**Reviewer 1**: 1) *Relation to regression strategies within person:* In BCI applications, samples are epochs from a subject. The focus is on linear signal changes (evoked response) or the powers (induced response). Our method could be applied in that context, with 'subjects' replaced by 'epochs' and a constant mixing matrix across subjects (leadfield from individual anatomy). In fact, the Riemannian methods were historically developed there. We will include a sentence in the paper.

2) *Include block diagram:* This a good idea. We will include one summarizing the different computations.

3 & 5) *Missing comparisons:* Extensions to other pipelines are interesting and we have computed them (See figure on the right). Y-axis depicts spatial filtering. "Identity" denotes its absence (using 'Ledoit-Wolf'-regularized covariance for geometric) and "random" for arbitrary spatial filters. We will detail a few points: i) Using supervised spatial filtering (aka SPoC) is critical to the log. diag. method but has little effect on the geometric method (expected due to the affine invariance). ii) The Riemannian method using Wasserstein distance does not need spatial filtering as it directly applies to rank-deficient matrices.

4) *significance levels:* Rejecting a null-hypothesis that differences are due to chance would require several independent datasets. Instead, we computed uncertainty estimates of paired differences using 100 Monte Carlo splits, 10% test set size. The rate of improvements in data samplings over the id. log. diag. baseline (in Fig. 3) is quantified by the percentage of splits with lower errors: sup. log. diag. 73%, id. Wass. 85%, unsup. geom. 96%, biophys. 95%. This suggests that improvements over the baseline will carry over to new data. The results will be included in the text.

6) *discussion on limitations of the work:* Primarily, our method falls behind the gold-standard based on biophysics (See Fig. 3). Another limitation is the low interpretability of the Riemannian methods, whereas inspection is readily available for other methods (See Fig. 4). We will clarify these 2 points in the discussion.

**Reviewer 2**: *More evaluations of geometric distance on Cam-CAN*: please see response 3) to Rev. 1.

*More evaluations on toy data*: To test a *suboptimal* scenario on toy data, we simulate data where each subject has a mixing matrix deviating from a reference ($A_i = A + E_i$ where entries of $E_i$ are i.i.d. $\mathcal{N}(0, \sigma^2)$). We also evaluate the interaction between the spatial filters and the projection method (See figure on the right). This shows: i) only when $\sigma = 0$, supervised + log. diag. and Riemann reach perfect performance, ii) else, geometric Riemann is uniformly better, and indifferent to filtering iii) Wasserstein, despite model mismatch, outperforms supervised + log. diag. with high $\sigma$. We will add this figure in the paper.

*Authors fail to discuss the basis of the different generative models:* We will add a sentence and references showing that log linearity relationships are standard for such brain signals [7 eq. (2)] or [Buzsaki & Mizuseki, Nat Neuro Rev] for evidence of log-linear links between signal power and cognition. Square root linearity is not commonly used for M/EEG modeling. It is included for its formal relationship with Wasserstein metric that can naturally cope with rank reduced data. We found it interesting that this Riemannian metric is consistent with this particular generative model.

*Model inspection results unnecessary:* We will replace it with block diagram asked by Rev.1 and move Fig. 4 to appendix.

*Missing references to recent works:* We will include the suggested references and nuance the claims in the discussion.

*Sections 2.1 and 2.2 are dense* We will do our best to reword those parts. In particular we will complement them with a more intuitive introduction to manifolds that we developed over the past months when interacting with practitioners.

*No code was provided*: Code for toy simulations was actually provided in the supplementary materials. Code for Cam-CAN methods and experiments will be fully released upon paper's acceptance.

*This paper introduces the first benchmark on MEG-based age prediction for the Cam-CAN dataset* Thanks for this encouraging comment. Our work was motivated by open issues in clinical neuroscience research and we are currently working together with medical doctors on translating the proposed method into the clinic.

**Reviewer 3**: *The discussion of pros and cons of the method could be increased:* see response 6) to Rev.1.

*A leave one-subject out cross-validation would have made a more realistic assessment of the performance* Results from neuroscience suggest that leave-one-out CV (LOO) may yield too optimistic scores (Varoquaux et al. 2016, NIMG, Woo et al. Nat Neuro Rev). Leave-one-subject-out should be helpful with multiple samples from each subject. Applying LOO, we find lower errors but higher variance: id. + log. diag.: 9.11 +/- 7.00, sup. +log. diag.: 8.65 +/- 6.72, id. + Wass.: 8.29 +/- 6.63, unsup. geom.: 8.02 +/- 6.02. We will include the results in the appendix.

[Meta-Review · NeurIPS 2019]

This paper presents a way to perform regression on a Riemannian manifold using rank-deficient covariance matrices. This is a novel approach to a new problem of age prediction from resting-state MEG. Reviewers agree this is a worthwhile contribution but have many suggestions for improvements that the authors are advised to consider.